# Coenzyme Q10 Efficacy Test for Human Skin Equivalents Using a Pumpless Skin-On-A-Chip System

**DOI:** 10.3390/ijms21228475

**Published:** 2020-11-11

**Authors:** Jisue Kim, Kyunghee Kim, Gun Yong Sung

**Affiliations:** 1Cooperative Course of Nano-Medical Device Engineering, Graduate School, Hallym University, Chuncheon 24252, Korea; prtty_u5588@naver.com (J.K.); seoulhee92@naver.com (K.K.); 2Integrative Materials Research Institute, Hallym University, Chuncheon 24252, Korea; 3Major in Materials Science and Engineering, School of Future Convergence, Hallym University, Chuncheon 24252, Korea

**Keywords:** pumpless skin-on-a-chip, 3-D culture, coenzyme Q10, human skin equivalents, skin anti-aging

## Abstract

A human skin equivalent (HSE) composed of the epidermis and dermis is cultured using a pumpless skin-on-a-chip system to supply cultures the desired flow rate using gravity flow without a pump or an external tube connection. Coenzyme Q10 efficacy is tested by adjusting its concentration, as it is known to have anti-aging and antioxidant effects in culture solutions. The relationship between the contraction rate of a full-thickness human skin equivalent and secreted transforming growth factor (TGF) β-1 is analyzed via enzyme-linked immunosorbent assay (ELISA). Following hematoxylin and eosin (H&E) staining, an image of the skin equivalent is analyzed to measure the epidermal layer’s thickness. The cell density and differentiation of the dermis layer are investigated. Gene and protein expression in the dermal and epidermal layers are quantitatively analyzed using quantitative real time polymerase chain reaction (qPCR) and immunohistochemical staining. As the coenzyme Q10 treatment concentration increased, the number of cells per unit area and the thickness of the epidermal layer increased, the expression level of filaggrin increased, and the contraction rate of full-thickness HSE was proportional to the amount of TGF β-1 secreted.

## 1. Introduction

The skin is the largest organ of the human body and is responsible for retaining water and protecting the body from external physical, chemical, and biological factors. The skin is roughly divided into an epidermal layer and a dermal layer. The epidermal layer is divided into the stratum corneum, the stratum lucidum, the stratum granulosum, the stratum spinosum, and the stratum basale, which form the outermost layer. The most important physical barrier to the invasion of external substances is the epidermis, consisting of the stratum corneum and keratinocytes. The stratum corneum, the uppermost layer of the epidermis, plays the most crucial role in the skin barrier function and because of this, damage to the subcutaneous tissue layer and the upper stratum corneum below that layer can be life-threatening. However, the physical barrier function is also performed on the skin, excluding the stratum corneum. Keratinocytes, which are the epidermal layer’s primary cells, move to the stratum corneum via the stratum spinosum and the stratum granulosum while dividing but this usually takes approximately four weeks. In the skin’s epidermal layer, the formed epidermal cells are repeatedly differentiated and detached to maintain homeostasis [1,2]. When such a barrier function of the skin is impaired, a skin disease appears. However, the water content of the skin barrier prevents the cracking of atopic skin, so that the barrier function of the skin is impaired. In this case, the skin is more likely to develop a disease, such as atopic dermatitis [3].

The skin barrier is maintained by various interactions between fibroblasts (FB), which constitute the dermis, and keratinocytes (KC), comprising the epidermis. In this process, secretions, such as interleukin (IL), keratinocyte growth factor (KGF), and transforming growth factor (TGF) generated in FB and KC, regulate these interactions [4]. Substances passed from the KC initiate the production of FB growth factors, FB proliferate, and the resulting secretions are transferred to the KC, causing KC proliferation and differentiation. As illustrated in Figure 1, KC-derived IL-1 and activation protein AP-1 were transferred to FB, which produced KGF, TGF, and Myb (myeloblastosis) in FB. Secretions, such as granulocyte-macrophage colony-stimulating factor (GM-CSF), are transferred to KC again and KC proliferation begins [4]. Of these, Myb, TGF β-1, and tumor necrosis factor (TNF) α-1 are factors that affect the formation of collagen in FB. Of these, TGF β-1 will maintain homeostasis under the regulation of KC [5]. Damage to the skin barrier leads to increased cytokine expression, which plays a vital role in restoring the skin barrier. However, the barrier damage persists and cytokine expression remains high. When the damage becomes persistent, inflammation occurs [6,7].

Previous studies have confirmed that TGF from incision wounds is predominantly induced in KC and inhibition of TGF plays a role in reducing injuries, resulting in TGF promoting collagen formation. Conversely, accelerated healing from an incised wound, when TNF signaling is blocked, also confirmed that collagen formation is promoted when TNF is suppressed [5]. A theory has been raised that wound healing is similar to the skin healing process following UV exposure—both wound healing and skin healing share mechanisms that include cell migration and proliferation. Skin deterioration due to UV light exposure takes on a form similar to skin aging [8].

Skin aging is divided into photoaging caused by UV light exposure and endogenous aging, occurring in relatively low-light exposure areas. Characteristics of endogenous aging include xeroderma, fine wrinkles, and decreased elasticity but photoaging is characterized by slightly deeper wrinkles and more irregular pigmentation than intrinsic aging. In addition, the thickness of the epidermal layer is reduced and the area of contact between the dermis and the epidermis is diminished. In the dermal layer, features can be found in which the thickness decreases with age and cells and blood vessels decrease overall [9].

Animal clinical trials are costly and time-consuming and tend to be banned due to various problems associated with animal welfare. Extensive studies have been conducted to generate human skin tissue models using a skin-on-a-chip system to overcome these limitations. [10] While in vitro 2D skin models are physiologically relevant, because real human skin tissue comprises various cell types, including keratinocytes, fibroblasts, vascular cells, and immune cells that can only be cultured in a 3D tissue environment, 2D models are insufficient. A 3D skin equivalent is superior to the 2D model in that it can mimic the 3D structure of the epidermis and dermis to create environmental conditions similar to human skin [5]. Recently, research on skin-on-a-chip constructs using a gravity flow system without a pump and tube to supply culture medium has also been conducted [11,12,13].

This study used a pumpless skin-on-a-chip, which provides the same physiological conditions as human skin. We systematically investigated the efficacy of coenzyme Q10 (CoQ10), which is well-known as an anti-aging agent and antioxidant. By adding CoQ10 to the culture medium with a pumpless skin-on-a-chip, the epidermis’ morphology and thickness were analyzed using H&E staining. The expressed proteins filaggrin, fibronectin, keratin 10, and involucrin were quantitatively analyzed using immunochemical staining. The gene expression of these proteins was also investigated using qPCR. The correlation between the human skin equivalent (HSE) contraction rate and the amount of TGF β-1 secreted during co-culture was also analyzed.

## 2. Results

### 2.1. Contraction of 3D Co-Cultured Human Skin Equivalents by a Pumpless Skin-On-A-Chip

Figure 2 and Figure 3 show changes in the contraction rate with the number of days of culture. There were almost no differences in the tendency of the contraction rate to increase depending on the concentration of coenzyme Q10. As shown in Figure 4, the increased contraction rate with culture days tended to increase with TGF β-1 secretion when the results of the TGF β-1 assay and the results of the contraction rate were compared. Shikano et al. reported that TGF β-1 regulates the amount of collagen I secreted by FB and the amount of TGF β-1 secreted is regulated by KC [5]. In this experiment, it was confirmed that the amount of secreted TGF β-1 increased as the culture period increased regardless of adding coenzyme Q10. It is thought that coenzyme Q10 is not directly involved in the contraction rate of HSE. However, the increased TGF β-1 secretion due to the subsequent paracrine signaling between KC and FB promotes collagen formation in the dermal layer, resulting in an increased contraction rate.

### 2.2. Changes in the Epidermal Layer Thickness and the Number of Fibroblast Cells

As shown in the H&E stained images in Figure 5, variation of the epidermal layer’s thickness was observed. In Figure 6, the epidermis’ thickness was measured in H&E images and compared quantitatively. Although the thickness did not proportionally increase as the concentration of coenzyme Q10 treatment increased, we confirmed that following treatment with a 10 µM of coenzyme Q10, the epidermis’ thickness was higher than that of the untreated samples. As shown in Figure 7, we confirmed that the number of fibroblasts increased concentration-dependently, compared with no treatment. These results were consistent with Žmitek et al.’s report that coenzyme Q10 induced the proliferation of skin fibroblasts and accelerated the production of epidermal basement membrane components [14].

### 2.3. Results of Protein Gene Expression Analysis

We analyzed four genes using qPCR. Collagen1A1 plays a role in strengthening and supporting tissues, including the skin and cartilage. Involucrin contributes to the extracellular shell that protects keratinocytes [15]. Filaggrin is present in the stratum granulosum and acts as an adhesive that aggregates keratinocytes [16]. Keratin 10 constitutes the cytoskeleton of epithelial cells [17]. As shown in the PCR results in Figure 8, the expression of protein genes increased with coenzyme Q10 concentration. Collagen I expression levels gradually increased with air exposure, concentrations increased, and keratin 10 and involucrin increased slightly. From these findings, we determined that increased concentrations of coenzyme Q10 strengthens skin tissues.

### 2.4. Results of Expressed Protein Immunohistochemistry (IHC) Staining

After capturing photomicrographs of IHC staining shown in Figure 9, protein expression levels were quantitatively analyzed using the Image J program in Figure 10. The protein expression levels were increased following treatment compared with untreated controls. According to Nemes and Steinert, filaggrin is a protein involved in the adhesion between keratinocytes. The weight of keratin and filaggrin account for 80–90% of the protein weight in the epidermis. It is also known that keratin is usually condensed and arranged through interaction with filaggrin in the final stage of epidermal differentiation [17]. The higher the expression level of filaggrin, the more congested and bundled the keratin is, strengthening the stratum corneum [18,19]. In skin diseases such as atopic dermatitis, low filaggrin values are observed in affected areas. Our IHC results show that the higher coenzyme Q10 treatment concentrations correlated with higher filaggrin expression in the stratum corneum containing KC.

Based on the results in Figure 9, we predicted that coenzyme Q10 affected the transfer of profilaggrin in the stratum granulosum to the stratum corneum to yield filaggrin through dephosphorylation and protein differentiation. E. Proksch et al. reported that involucrin, a marker of the stratum corneum’s degree of differentiation, initiates cell differentiation. The inner stratum corneum of the cell membrane plays the most crucial role in performing the skin’s barrier function. If there is a problem with the skin barrier, the level will be lower if normal differentiation does not occur [20]. This study confirmed increased protein expression following high concentration coenzyme Q10 treatments. In addition, Elias et al. [21] reported that involucrin constitutes the keratinocyte membrane and is the outer shell that protects keratinocytes. By analogy, regarding the content of proteins that contribute to keratinocytes, the skin barrier is more effective as the concentration of coenzyme Q10 increases. No significant difference was found between CoQ10 treatment and fibronectin expression in the dermal layer.

## 3. Discussion

Because it protects the skin from ROS and reduces UV-induced inflammatory responses, coenzyme Q10 plays an essential role in intracellular anti-aging and antioxidant activity, helping to reduce vitamin E. HSEs were cultured using a pumpless skin-on-a-chip and the concentration of coenzyme Q10 was adjusted in the culture medium to test its efficacy. Coenzyme Q10 is not directly involved in the contraction rate of HSE and the increase in TGF β-1 secretion is due to the continued paracrine signaling between KC and FB, which promotes collagen formation in the dermal layer and increases the contraction rate. As the concentration of coenzyme Q10 increased, the epidermal layer’s thickness increased and fibroblasts also increased. These results matched well with the report that coenzyme Q10 induces the proliferation of skin fibroblasts and accelerates the production of epidermal basement membrane components. Higher coenzyme Q10 treatment concentrations induced increased expression of genes and proteins, such as filaggrin, fibronectin, keratin 10, and involucrin. When the pumpless skin-on-a-chip is used, compared to 2D culture, the epidermal layer and the dermal layer can be co-cultured in 3D, so the efficacy of the drug is analyzed histologically and molecular biologically with a sample with greatly improved similarity to human skin. It is expected to be very useful in future drug efficacy tests. We predict this system can be utilized as a useful disease model and an alternative to animal testing methods during cosmetics development.

## 4. Materials and Methods

### 4.1. Skin-On-A-Chip

A skin chip was produced by mixing polydimethylsiloxane (PDMS) and a curing agent at a ratio of 10:1. The skin chip is composed of a lower part, an upper portion and a porous membrane (24 mm Transwell with 0.4 μm pores/Corning) is located between the lower chip and the upper chip. There is an 8 mm cylindrical chamber in the center of the upper part and cells are cultured in the chamber on the membrane. When filled in both badge chambers, the badges were allowed to circulate through the lower channel (width: 200 μm, height: 150 μm).

### 4.2. Cell Culture

Rat tail collagen (Corning Inc., New York, NJ, USA) and human dermal fibroblasts (Biosolution Co., Ltd., Seoul, Korea, catalog No. MC1233, Adult) were mixed and gelled in an incubator and then cultured in DMEM(Lonza, Dulbecco Modified Eagle Medium) medium at 37 °C, in a 5% CO_2_ incubator for 5 days. After forming the dermal layer for 5 days, human epidermal keratinocytes (Biosolution Co., Ltd., Seoul, Korea, catalog No. MC1323, Adult/Foreskin) were attached and the ball culture was cultured for 2 days using KGM (Lonza) medium. After that, an E-Media badge was used in the air exposure stage, where cells were directly exposed to the air for differentiation. E-media composition: DMEM/Ham’s F12 (EGF-1 10 ng/mL, hydrocortisone 0.4 μg/mL, insulin 5 μg/mL, transferrin 5 μg/mL, 3,3,5-triiodo-L-thyonine sodium salt 2 × 10^−11^ M, cholera toxin 10^−10^ M, 10% (*v*/*v*) FBS, and 1% penicillin/streptomycin. Coenzyme Q10 treatment was performed at the air exposure stage.

### 4.3. Drug Treatment

Coenzyme Q10 and poloxamer 407 were mixed in a 1:5 ratio (0.0086334 g: 0.43167 g) and dissolved in 95% ethanol [22]. Then, the mixture was melted in an oven at 70 °C., cooled to room temperature, mixed for 15 min in a light half period, and then mixed with E-Media for treatment. Based on 50 mL of E-Media solution, 0.1 μM, 1 μM, and 10 μM were prepared and used.

### 4.4. The Gravity Flow System

A gravity flow system was used to efficiently supply the culture medium to 3D cell cultures [14,23,24]. The culture solution flows in the direction of gravity. This device is divided into a computer, a motor, and a stage and the angle and interval time can be controlled via a computer system. It works by rocking both sides at an angle of 15 degrees. The tilting causes the medium to circulate through the microfluidic channel as shown in Figure 11.

### 4.5. ELISA

Badges were extracted and ELISA analysis was performed to confirm the expression level of TGF β-1 in the artificial skin equivalent. For air exposure, the medium was removed 3, 5, and 7 days after exposure. Experiments were performed using a human TGF β-1 Sandwich ELISA Kit (LSBIO, Seattle, WA, USA, LS-F2548-1, Sandwich ELISA) and a SpectraMax M2 spectrophotometer (Molecular Devices Inc., San Jose, CA, USA) at a wavelength of 450 nm. The OD values were measured for each sample.

### 4.6. The Number of Cells

The number of cells were counted from the H&E images in Figure 5 and converted into the number per unit area.

### 4.7. Quantitative Analysis by Real-Time PCR

For quantitative real-time polymerase chain reaction (qRT-PCR) analysis, mRNA was extracted from the tissue unit of the HSE sample using the TRIzol method. The extracted mRNA was quantified in SpectraMax M2 Microplate Readers (Molecular Devices Inc.). cDNA was synthesized by processing the previously extracted mRNA by amfiRivert cDNA Synthesis Platinum Master Mix (GenDEPOT, Barker, TX, USA). The qPCR was run using the LightCycler^®^ 480 SYBR Green I Master (Roche, Basel, Switzerland) in LightCycler^®^ 480 Instrument II (Roche, Basel, Switzerland). The following primer pairs were used; for human 18s rRNA gene, 5′-GGCGCCCCCTCGATGCTCTTAG-3′ and 5′-GCTCGGGCCTGCTTTGAACACTCT-3′; for human filaggrin gene, 5′-GGAGTCACGTGGCAGTCCTCACA-3′ and 5′-GGTGTCTAAACCCGGATTCACC-3′; for human involucrin gene, 5′-CCGCAAATGAAACAGCCAACTCC-3′ and 5′-GGATTCCTCATGCTGTTCCCAG-3′, for human keratin 10 gene, 5′-CCGGAGATGGTGGCCTTCTCTCT-3′ and 5′-GGCCTGATGTGAGTTGCCATGCT-3′. Human 18s rRNA gene was used as the housekeeping gene.

### 4.8. Image Analysis through H&E and Immunohistochemistry Staining

For hematoxylin and eosin (H&E) staining, fix the tissue sample with formaldehyde. Place a small amount of melted paraffin into the bottom of the tissue mold and push the tissue flat against the bottom of the box. After the paraffin hardens, remove the mold/box. The prepared paraffin tissue block is cooled to 0–4 °C and the paraffin-embedded tissues are sectioned into 4 µm thickness using a microtome. Before reacting with aqueous reagents, deparaffinize and rehydrate the sections by immersing them successively for agitation in xylene, 100% ethanol, and 70% ethanol, followed in an appropriate buffer. Sections were stained with eosin solution (Sigma-Aldrich Co., St. Louis, MO, USA) for 1 min. After washing with distilled water, sections were dehydrated with an alcohol wash and unstained with xylene. The stained tissue was observed using an inverted optical microscope (Olympus, DP73, Tokyo, Japan).

For immunohistochemistry (IHC) staining, incubate the deparaffinized paraffin-embedded sections in primary antibodies specific to fibronectin (ab2413), involucrin (ab53112), cytokeratin (ab76318), and filaggrin (ab81468) separately overnight at 4 °C and process using Benchmark XT Auto-stainer System (Roche), following the manufacturer’s instructions. Include slides for appropriate controls. All antibodies were purchased from Abcam Inc, Cambridge, UK and used at 1:200 dilution. Incubate with horseradish peroxidase (HRP)-conjugated secondary antibody and visualized by 3,3-diaminobenzidine (DAB) to detect the protein of interest. Stain the nuclei with hematoxylin. The slides were scanned using an optical microscope (Olympus, IX73-F22PH) with attached optical camera (Olympus, DP73-ST-SET). Each sample was scanned 5 times and IHC image analysis was performed using the free software ImageJ Fiji (Java 1.8.0_172 64bit) program.

## Figures and Tables

**Figure 1 ijms-21-08475-f001:**
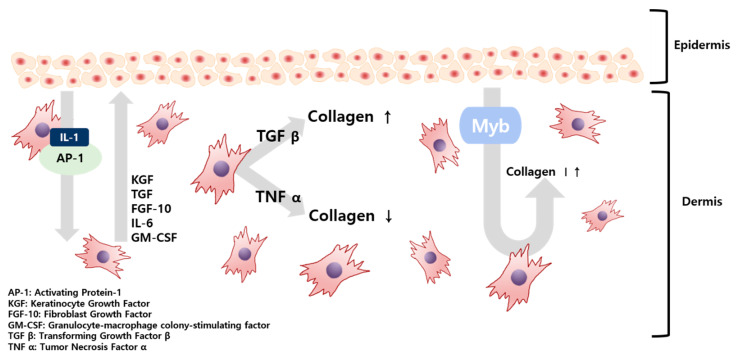
Schematic illustration of the paracrine signaling between fibroblasts and keratinocytes in the skin tissue.

**Figure 2 ijms-21-08475-f002:**
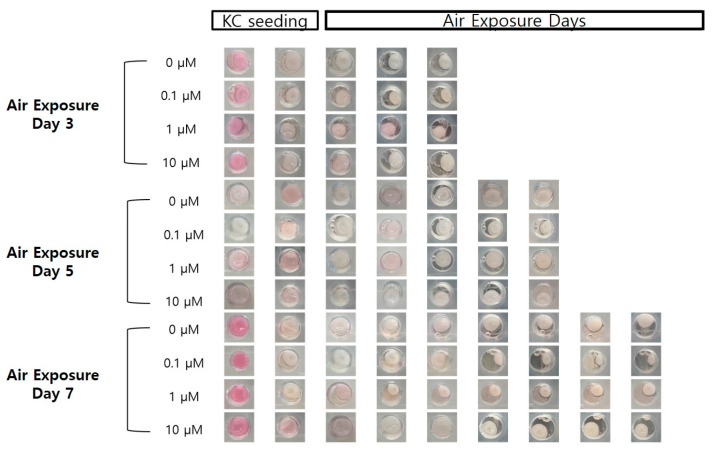
Photographic images of the samples, following air exposure, for the different concentrations of coenzyme Q10 (bar = 8 mm).

**Figure 3 ijms-21-08475-f003:**
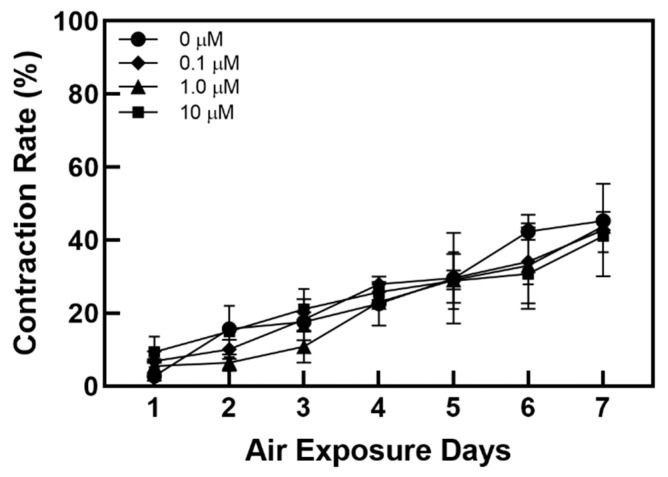
Variation of the contraction rate of the samples as a function of the air exposure period with the four different concentrations of coenzyme Q10.

**Figure 4 ijms-21-08475-f004:**
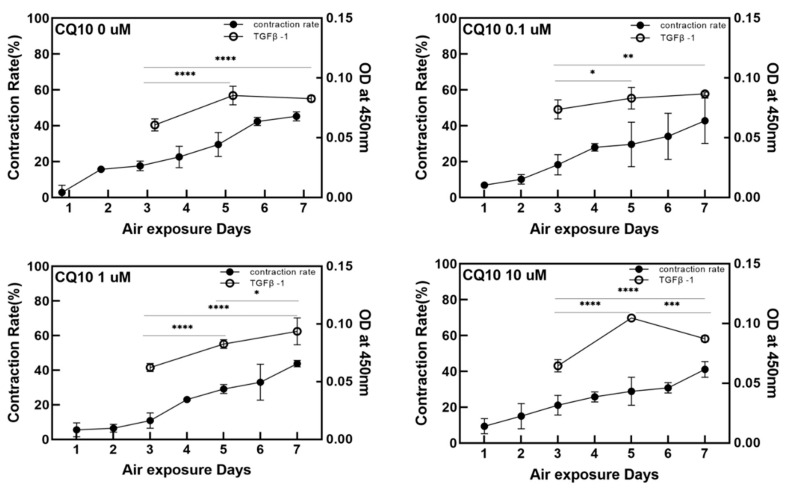
Variation of the contraction rate of the samples (left *Y*-axis) and the relative optical intensity of the transforming growth factor (TGF) β-1 from enzyme-linked immunosorbent assay in the culture media as a function of the air exposure period (right *Y*-axis) with the four different concentrations of coenzyme Q10. (*p*-value: * < 0.05, ** < 0.01, *** < 0.001, and **** < 0.0001).

**Figure 5 ijms-21-08475-f005:**
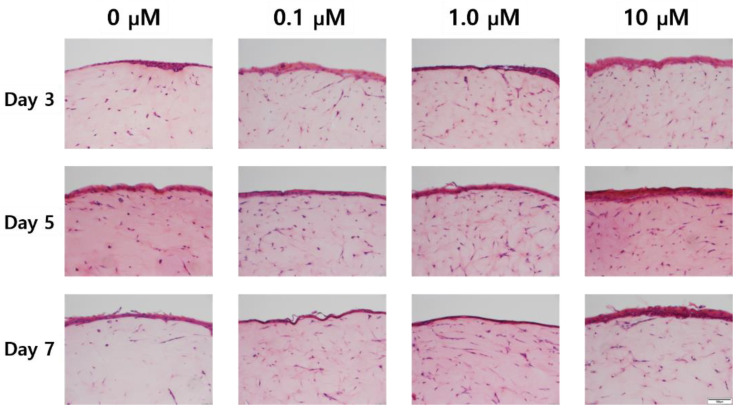
Hematoxylin & eosin staining images of the sample with air exposure and coenzyme Q10 concentration (bar = 100 μm).

**Figure 6 ijms-21-08475-f006:**
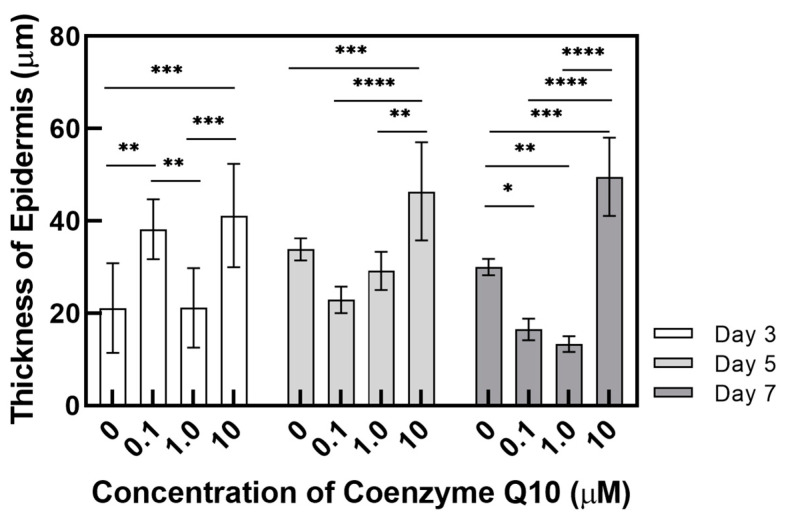
Variation of the thickness of the epidermal layer as a function of the coenzyme Q10 concentration after 3 days, 5 days, and 7 days of air exposure (*p*-value: * < 0.05, ** < 0.01, *** < 0.001, and **** < 0.0001).

**Figure 7 ijms-21-08475-f007:**
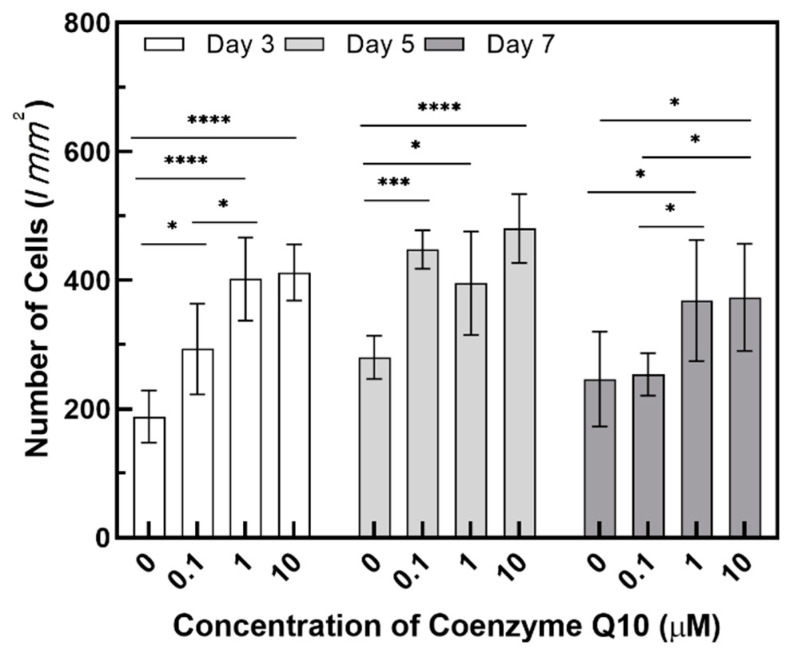
Variation of the number of fibroblasts in the dermal layer as a function of the coenzyme Q10 concentration after 3 days, 5 days, and 7 days of air exposure (*p*-value: * < 0.05, *** < 0.001, and **** < 0.0001).

**Figure 8 ijms-21-08475-f008:**
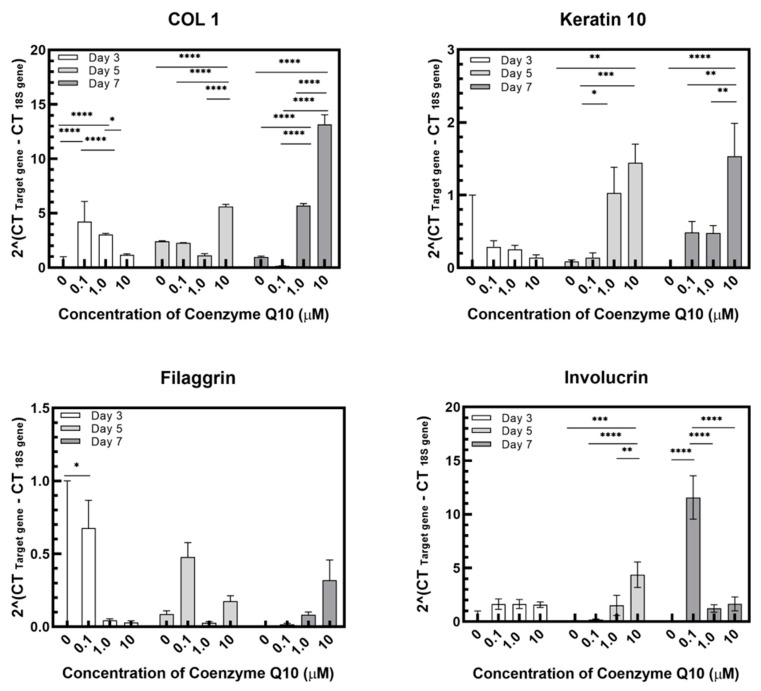
Relative gene expression of collagen 1, keratin 10, filaggrin, and involucrin in samples as a function of the coenzyme Q10 concentration after 3 days, 5 days, and 7 days of air exposure (*p*-value: * < 0.05, ** < 0.01, *** < 0.001, and **** < 0.0001).

**Figure 9 ijms-21-08475-f009:**
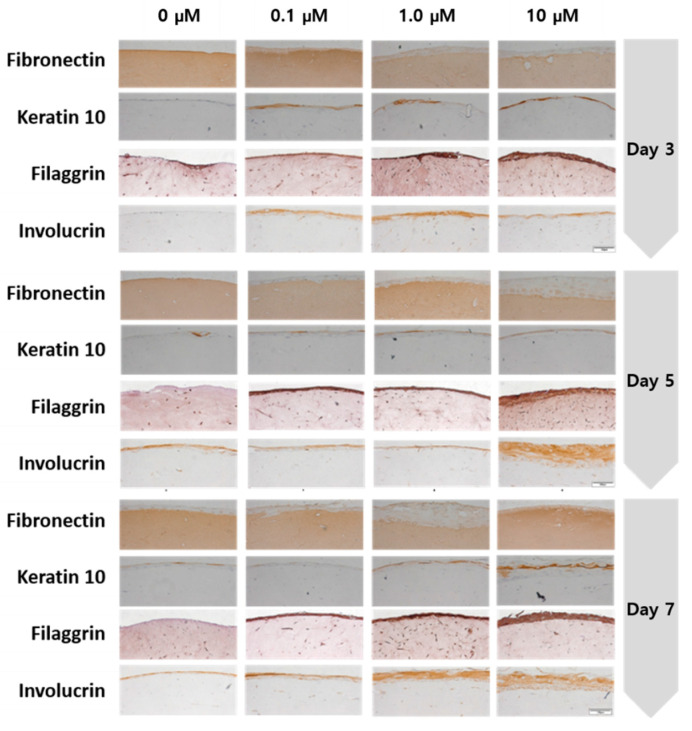
Immunohistochemistry stained images of the fibronectin, keratin 10, filaggrin, and involucrin for the sample as a function of the coenzyme Q10 concentration following 3 days, 5 days, and 7 days of air exposure (bar = 100 μm).

**Figure 10 ijms-21-08475-f010:**
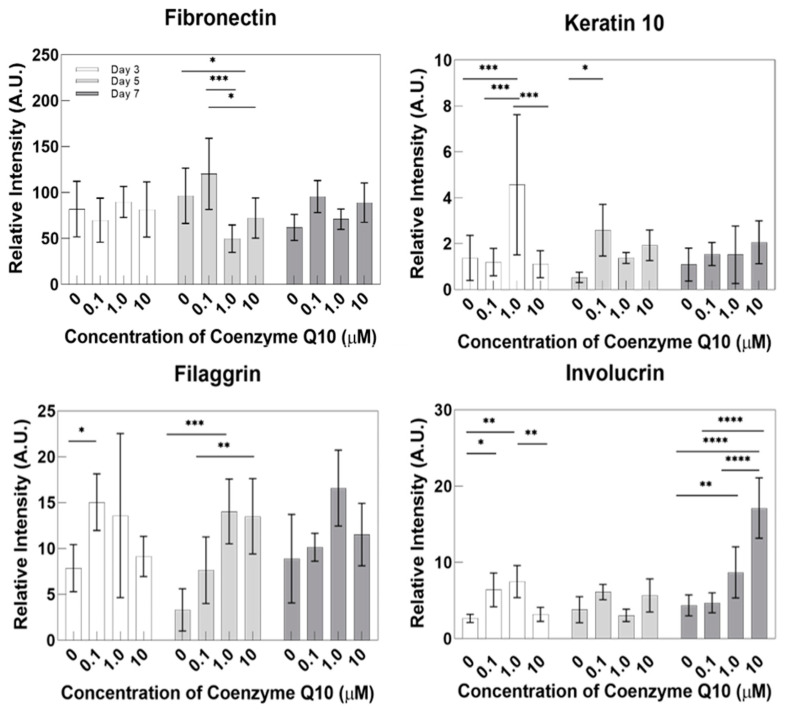
Quantitative analysis of protein expression for fibronectin, keratin 10, filaggrin, and involucrin from the immunohistochemistry stained images (*p*-value: * < 0.05, ** < 0.01, *** < 0.001, and **** < 0.0001).

**Figure 11 ijms-21-08475-f011:**
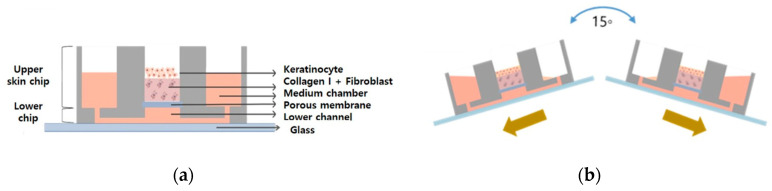
Schematic illustration of (**a**) the pumpless-skin-on-a-chip and (**b**) gravity flow directions by using the gravity flow system.

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
