# Peer review of "Coenzyme Q10 Efficacy Test for Human Skin Equivalents Using a Pumpless Skin-On-A-Chip System"

_ijms, 2020, doi:10.3390/ijms21228475_

Round 1

Reviewer 1 Report

This study demonstrates the capability of the skin-on-a-chip system for the efficacy evaluation of anti-aging materials. For this purpose, the authors utilized a pumpless 3D skin models by culturing dermal fibroblasts and keratinocytes as a layered form and treated the coenzyme Q10 as an anti-aging drug. The morphological features such as the thickness epidermis and proliferation of fibroblasts showed direct evidence of anti-aging. Further, the changes in gene expression related to the skin functions supported the results of histological analysis. The data present in this study is solid and sufficiently support the strong potential of skin-on-a-chip as a testing platform. This manuscript is well-organized and well-written. Therefore, the reviewer suggests acceptance of this manuscript after addressing the minor issue.

The title can be simplified for the delivery of the main idea. Especially, the meaning of the ‘human skin equivalents’ and the ‘skin-on-a-chip’ are overlapped.

Overall, the quantity data (number of images) are sufficiently large, and the quality was also quite good to support their idea.

Author Response

Thank you for your kind suggestion. But we think that the title is more comprehensive for the readers of this journal.

Reviewer 2 Report

This article is a verification of the in vitro system using organ-on-a-chip to overcome the limitations of existing animal testing and clinical applications using Coenzyme Q10. Although it is considered highly efficient and very beneficial to researchers considering experiments in the same way as the text in the future, it is insufficient to adopt at the current level, so it is expressed as follows:

[Major point]
1. Despite being a dermatological approach using a chip from an engineering perspective, the contents of the integration were written very comprehensively. It is necessary to write more implicitly with key points.

2. Since there is no picture of the most important pumpless organ-on-a-chip in the text, it is not known how the original organ task and related data of the text were obtained, so include the relevant pictures and schematics.

3. The overall result description is not orderly, although it refers to Figure number.

4. The amount of TGF-b1 has been noted to have increased, but it cannot be determined without comparison of statistical significance.

5. Comparing the Thickness value of Figure 7, the variation was announced on the same day, but there is no dependency on Co-Q10 concentration .

6. How did you measure the number of cells in Figure 8? In some cases, including the control case, the increased number may have decreased, please explain.

7. The results of the chip system validity mentioned in the article through Co-q10 are understood, but without controls such as 2D cultured cells or tissues, this system alone is not sufficient to appeal the usefulness and effectiveness of the results. Please express the author's opinion on this part.

[Minor point]
- Please explain clearly in the manuscript M&M whether gene expression is seen in tissue or cell separation.

- Write clearly the source and specification of each material, including cells.

Author Response

We appreciate the thorough review of our manuscript. We revised the manuscript extensively for clarity according to the Reviewer's comments.

Reviewer 2

This article is a verification of the in vitro system using organ-on-a-chip to overcome the limitations of existing animal testing and clinical applications using Coenzyme Q10. Although it is considered highly efficient and very beneficial to researchers considering experiments in the same way as the text in the future, it is insufficient to adopt at the current level, so it is expressed as follows:

[Major point]
1. Despite being a dermatological approach using a chip from an engineering perspective, the contents of the integration were written very comprehensively. It is necessary to write more implicitly with key points.

: According to reviewer’s comment, we deleted the basic introduction of CoQ10 materials (line 79 ~ 115 of the original submission) in “1. Introduction” section.

  1. Since there is no picture of the most important pumpless organ-on-a-chip in the text, it is not known how the original organ task and related data of the text were obtained, so include the relevant pictures and schematics.

: We added the figure showing detailed chip structure and gravity flow motion in Fig 11(line 329~333 of the revised manuscript).

  1. The overall result description is not orderly, although it refers to Figure number.

: Our results were arranged in order of contraction of HSE, H&E result, PCR, and IHC. And section title of  2-1 and 2-2 revised into “2-1. Contraction of 3D co-cultured human skin equivalents by a pumpless skin-on-a-chip”(line 98) and “2-2. Changes in the epidermal layer thickness and the number of fibroblast cells”(line 192), In addition, M&M subsections were rearranged according to results.  

  1. The amount of TGF-b1 has been noted to have increased, but it cannot be determined without comparison of statistical significance.

: We added the p-values into the Fig.4 (line 187).

  1. Comparing the Thickness value of Figure 7, the variation was announced on the same day, but there is no dependency on Co-Q10 concentration .

: Thank you for the sharp point. We revised the paragraph into “As shown in the H & E stained images in Fig. 5, variation of the epidermal layer's thickness was observed . In Fig. 6, the epidermis' thickness was measured in H & E images and compared quantitatively. Although the thickness did not proportionally increased as the concentration of coenzyme Q10 treatment increased, we confirmed that following treatment with a 10µM of coenzyme Q10, the epidermis' thickness was higher than that of the untreated samples.” (line 193~197)

  1. How did you measure the number of cells in Figure 8? In some cases, including the control case, the increased number may have decreased, please explain.

: We added the measurement of the number of cells into M&M section.(line 342)  As shown in Fig. 7 in revised manuscript, the sample with day 5 and 1 uM CoQ10 only did not increase, but it shows almost similar values within the error bar, so I think it does not deviate significantly from the overall trend.

  1. The results of the chip system validity mentioned in the article through Co-q10 are understood, but without controls such as 2D cultured cells or tissues, this system alone is not sufficient to appeal the usefulness and effectiveness of the results. Please express the author's opinion on this part.

: We have added the following sentence into the end of conclusion. “When pumpless skin-on-a-chip is used, compared to 2D culture, the epidermal layer and the dermal layer can be co-cultured in 3D, so the efficacy of the drug is analyzed histologically and molecular biology with a sample with greatly improved similarity to human skin. It is expected to be very useful in future drug efficacy tests.”(line 403~406)

 [Minor point]
- Please explain clearly in the manuscript M&M whether gene expression is seen in tissue or cell separation.

: We revised the first sentence of “3-7. Quantitative analysis by Real-time PCR” as follows: “For quantitative real-time polymerase chain reaction (qRT-PCR) analysis, mRNA was extracted from the tissue unit of the HSE sample using the TRIzol method.” (line 348 ~ 349)

- Write clearly the source and specification of each material, including cells.

: We added the source and specification of each material, including cells as follows: “Rat tail collagen (Corning Inc., New York, USA) and human dermal fibroblasts (Biosolution Co., Ltd., Seoul, Rep. of Korea, catalog No. MC1233, Adult) were mixed and gelled in an incubator, and then cultured in DMEM medium at 37° C. in a 5% CO2 incubator for 5 days. After forming the dermal layer for 5 days, human epidermal keratinocytes (Biosolution Co., Ltd., Seoul, Rep. of Korea, catalog No. MC1323, Adult/Foreskin) were attached, and the ball culture was cultured for 2 days using KGM (Lonza) medium.”(line 297 ~ 302)

Round 2

Reviewer 2 Report

No more comments.